# How the COVID-19 Pandemic Affects Risk Awareness in Dentists: A Scoping Review

**DOI:** 10.3390/ijerph19094971

**Published:** 2022-04-20

**Authors:** Thomas Gerhard Wolf, Leonardo De Col, Seyed Ahmad Banihashem Rad, Paolo Castiglia, Antonella Arghittu, Mina Cannavale, Guglielmo Campus

**Affiliations:** 1Department of Restorative, Preventive and Pediatric Dentistry, University of Bern, Freiburgstrasse 7, 3012 Bern, Switzerland; thomas.wolf@zmk.unibe.ch (T.G.W.); leonardo.decol@students.unibe.ch (L.D.C.); ahmad.banihashem@yahoo.com (S.A.B.R.); 2Department of Periodontology and Operative Dentistry, University Medical Center of the Johannes Gutenberg-University Mainz, 55131 Mainz, Germany; 3Department of Surgery, Microsurgery and Medicine Sciences, School of Dentistry, University of Sassari, Viale San Pietro 3/c, 07100 Sassari, Italy; castigli@uniss.it (P.C.); arghittu.antonella@gmail.com (A.A.); 4Direzione Igiene e Controllo delle Infezioni Ospedaliere, University Hospital of Sassari, Via Padre Manzella 4, 07100 Sassari, Italy; 5Independent Researcher, 81030 Casaluce, Italy; minacannavale29@gmail.com

**Keywords:** dentistry, dentist, COVID-19, review, risk awareness

## Abstract

Dentists are on the frontline of infection, especially when it comes to respiratory viruses like the new coronavirus. The purpose of this paper is to conduct a scoping review to better understand dentists’ risk awareness, awareness of COVID-19 symptoms, preventive measures, and effective methods of COVID-19 infection prevention and management. This paper systematically assesses the published literature on dentistry and COVID-19. Various electronic databases including Ovid MEDLINE, Scopus, Embase, and MEDLINE via PubMed were searched up to 9 September 2021. Overall, 39 papers were included. Almost the entirety of dentists (94.5%) reported awareness of the three most common COVID-19 symptoms, and a risk awareness score of about 90% was shown, while 88.2% of dentists reported adopting preventive measures. More than 50% did not want to treat infected people. While 70.3% of dentists recommended usage of N95 masks, the rate of dentists using them was below 40%. Sufficient awareness of risks during the pandemic was found in dentists. Although they were using preventive measures, there remains upside potential for adopting all recommended measures. Further, the usage of N95 masks is improvable, even though the benefit of wearing them could not be confirmed.

## 1. Introduction

Discovered on 31 December 2019 in Wuhan, the COVID-19-Outbreak was declared a pandemic by the WHO on 11 March 2020 [1,2].

Related to the pandemic, dental health care professionals are seen as a high-risk group. Dental healthcare workers are considered at the frontline in the current pandemic because they are susceptible to infection through various airborne and contact routes. Taking into consideration the worldwide spread of SARS-CoV-2, it is of utmost importance to provide standard feasible preventive strategies in dental settings, because dentists and other oral health practitioners, such as dental assistants and dental hygienists, are highly vulnerable due to direct contact with the respiratory aerosols of patients infected with the novel coronavirus [3,4,5].

For these reasons, dentists were advised to change their practice management to contain the spread of the virus and decrease the risk of infection for dentists and their patients [6,7,8].

At the same time, evidence suggests that healthcare workers’ accurate knowledge plays a vital role in outbreak management. In this regard, health education and continuous training activities designed to bring about cultural and behavioral changes in health professionals and the public, making the best use of scientific evidence, are of well-known importance [9]. However, misinformation flow during the pandemic exacerbated the situation for healthcare workers and policymakers [10]. The decision-making process is strongly influenced by what is read online, especially in relation to particular health issues currently widely discussed (e.g., the COVID-19 pandemic affects risk awareness and risk perceptions) [11].

Furthermore, in the current pandemic context, the social isolation imposed on the population has further contributed to an increase in the use of the web, exposing population cohorts to a greater amount of information. During the COVID-19 pandemic, repeated media exposure to news, discrepancies in the number of infections and mortality data reported in different countries, and conflicting media opinions on the urgency of a vaccine made it difficult for people to find reliable sources of information [12].

Healthcare professionals, who have a vital responsibility to their patients to ensure that those in need of medical care are treated, are not exempt from the influence of these determinants of risk awareness; this appears even more significant when one considers the fact that oral healthcare professionals could be a route of transmission of the virus to patients, colleagues or family members. Furthermore, if adequate safety precautions are not implemented, dental clinics are likely to expose patients to cross-contamination [13]. Shutting down dental clinics during the epidemic may reduce the number of people infected, but it would exacerbate the distress of those who require emergency dental treatment [14].

As a result, it is important to assess these personnel’s knowledge and understanding of the disease’s transmission and management mechanisms. Investigating the influence of the perception of a health risk on the adoption of the correct preventive measures is a topic of relevant importance for health surveillance and for the implementation of the best prevention strategies on the part of health and social service providers and policymakers. Many attempts have been made to provide clear and convenient instructions for managing dental patients and reduce risk to dentists and other oral health practitioners [13]. However, different countries published various, even contradictory, guidelines, a situation which was challenging for dentists [15,16].

Based on the above, this review aims to summarize dentists’ risk awareness (e.g., how dentists assessed the risks inherent to their profession), usage of preventive measures (e.g., common or advanced preventive measures in dental health care and usage of personal protective equipment), and treatment of infected people during the pandemic, as dentists must understand the risks during such an extraordinary situation. Furthermore, it aims to give recommendations to increase risk awareness in case of another pandemic. The current study’s findings should also aid decision-makers in formulating coronavirus management guidelines in dentistry.

## 2. Materials and Methods

This paper was designed as a scoping review. Several scientific electronic databases (Ovid MEDLINE, Scopus, Embase, MEDLINE via PubMED) up to 9 September 2021 were explored systematically by keyword (Table 1). The SPIDER [17] tool was used to build up the search strategy.

**S**(ample): “Dentistry” OR “dental” OR “dentist”

**P**(henomenon) of **I**(nterest): All the English-language articles that related to knowledge, attitude, and awareness and risk perception of oral health workers about contagious respiratory outbreaks.

**D**(esign): not restricted.

**E**(valuation): “attitude*” OR “Risk awareness*” OR “behavior change *” OR “risk perception“.

**R**(esearch type): not restricted.

The selection was performed using structured procedures. The titles and abstracts of all included papers were examined by two independent reviewers (L.D.C. and S.A.B.R.) in order to determine their relevance and whether they matched the planned inclusion criteria. The two authors read and scored all the papers at the eligibility stage. The inter-rater reliability, Cohen’s Kappa, was measured [18].

After examining the studies in their full texts, an expert was called in (G.C.) for the final selection of articles where the reviewers did not reach consensus.

The relevant data from the publications were collected using a data extraction form. The author, the region, the sample size, and the results found in the original articles were all included on this form. The new case increment observed during the different surveys from all the included papers was retrieved [19].

## 3. Results

Two reviewers (L.D.C., S.A.B.R.) screened the studies by title (Figure 1) after removal of duplicates (*n* = 518). After the title screening, 211 papers remained and were screened by abstract, of which 78 were left for full-text screening. The inter-rater reliability was quite good, with a score of 83.7%. Data extraction was then performed on 56 articles. After the data extraction, 39 studies were included. The studies excluded after the full-text review are listed in Appendix A, while the studies excluded after data extraction are given in Appendix A. The list of the studies included sorted by country is reported in Table 2. All the surveys were carried out during the first wave of the pandemic (Spring–Summer, 2020) and the majority during the expansion phase of the pandemic (32 surveys out of 39), according to the relative changes (%) COVID-19 cases. Data about COVID-19 cases in Lebanon were not retrieved.

### 3.1. Awareness of Symptoms and Risk Awareness

An overall agreement of 94.4% with a SD of 4.8% was achieved for the dentists’ awareness of COVID-19 symptoms (Table 3). Only the awareness of the three most common symptoms, fever, cough, and shortness of breath, was considered, showing a percentage over 80% in all the countries [20,26,27,28,34,35,39,40,42,49,50,55].

Several aspects linking to dentists’ risk awareness were mentioned (Table 4). Over 80% of dentists regarded the virus as fatal [26,58]. About three-quarters of dentists ranked the virus as moderately dangerous or dangerous [20,34,35,41,49,51]. More than two-thirds of dentists favored practice closure due to the potential fatality of the virus until a decline in COVID-19 cases was noticed [59]. About 70% of dentists assessed the dental practice as more dangerous than other engagements for spreading the virus [60]. In another review, only 40% of dentists thought that dental treatment could increase the spread of COVID-19, while especially older dentists denied it [56]. Around 90% of dentists saw their profession as one of the most endangered during the pandemic, as they work in close contact with patients and many treatments produce aerosols [21,23,26,29,30,35,45]. Whether COVID risk is higher for the dentist or the patients, the included studies showed contradictory answers, but more than half of the dentists believed it to be higher for dentists [35,36,60].

The overall risk awareness score exceeded 90% in all studies which evaluated the scores [31,43,46].

### 3.2. Taking Preventive Measures

As an indicator for the application of infection control measurements, preventive measures (Table 5) achieved a mean rate of 88.2% with an SD of 14.6% [22,23,25,37,40,41,47,49,57,58]. The lowest score of 54.8% was achieved in Pakistan [47]. All other studies besides except four [23,37,41,47] reported values over 80% relating to preventive measures.

### 3.3. Treatment of COVID-19 Affected Subjects

Almost 50% of dentists (Figure 2) were willing to treat affected subjects. Considerable differences in the number of dentists willing to treat infected people were present, ranging from 31% in Austria up to 84.4% in India [20,23,32,41,47,49,56,57,61]. The values were higher in central Asian countries; three different studies were carried out in India, and all reported different numbers; also, the highest rate was reported in India, where 84.4% of dentists said they would provide treatment to infected people [29,32,61].

### 3.4. Use of the N95 Mask

70.3% (SD 22.1%) of dentists considered it a good idea to wear N95 masks during the pandemic, while 37.7% (SD 27.6%) actually wore them (Table 6). Only in Europe, the Western Pacific, and Spain, did less than 50% of dentists think N95 masks should be worn [53,62]. The single country with the highest usage of these masks was Austria, where only 10.3% of dentists did not use them [20]. Despite the moderately high value of 67.8% in North America recommending the use of N95 masks, a small number of 16.4% used the masks [62].

## 4. Discussion

The purpose of this research was to report dentists’ risk awareness, awareness of the COVID-19 symptoms, preventive measures, and effective methods of preventing and managing the infection.

The data has shown an acceptable level of risk awareness. Most dentists were aware of being endangered due to their profession. Dentists were also well aware of the symptoms of COVID-19, which is very important, primarily to protect and decrease the spread of the virus by recognizing symptoms in potentially infected patients. The risk awareness scores could be connected to the high awareness of symptoms score, because most symptoms are related to the mouth and pharynx area, which is the working area for dentists. In order to protect themselves, dentists need to be aware of COVID-19 symptoms [49], which explains the high score for this issue.

Also, pooled mortality rate of 5.6% has been reported [63]. Dentists were right to assess the virus as dangerous or even fatal. In Pakistan, the lowest rate of preventive measures was reported [47]; only half of the respondents used preventive measures and were willing to treat infected people. In comparison to the study conducted in Brazil during a similar time frame [22], where the rate of preventive measures was at 98%, Pakistan had 4–17 times fewer confirmed COVID-19 cases per million [19]. The reason for this big difference could be the lower rate of cases, so the danger of the virus was less present than in, for example, Brazil. Moreover, the majority of participants in the studies conducted in Turkey [56] and Saudi Arabia [49] described the lowest level of willingness to treat infected people. It is worth mentioning that both studies were after the first wave of COVID-19 in these countries. At the time these studies were carried out, the case fatality rate exceeded the world’s average fatality rate by up to 4% [19], this may have boosted risk awareness in Turkey and Saudi Arabia, which explains the lower willingness to treat infected patients. Most of the studies in the sample relating to treatment of infected people took place in the first wave of COVID-19. Dentists’ opinions in the research above were in line with those of Jordanian dentists, where the majority of participants (82.6%) opted to avoid treating suspected COVID-19 patients despite the absence of signs or symptoms of the illness [38]. 

The use of preventive measures as routine infection control for every patient before the pandemic was also stated [38,52,59]. Different numbers were found when there was a change of infection control due to the virus. At the same time, a study from Lebanon [41] reported low adoption, studies from the UAE and Saudi Arabia reported a moderate percentage [52,57] and the highest percentage was found in Italy with 91.6% [34,38]. Of the mentioned countries, only Lebanon showed a value of confirmed COVID-19 cases below the world’s average in the study period [19]. Italy was one of the most affected countries during the beginning of the pandemic [34,35]; this could be the reason why the Italian study revealed higher change scores, probably they were more eager to counteract the virus spread. It is also thought that by raising risk awareness, the adoption of preventive measures can be enhanced [37].

The dentists in selected studies showed sufficient but developable usage of preventive measures and personal protective equipment. Preventive measures for infection control, including standard barrier protection with masks, gloves, and safety goggles and use of antiseptic techniques like hand-washing or sterilization of instruments were routinely undertaken before the pandemic [64,65]. Related to the pandemic, dental health care professions are seen as a high-risk group. Their work entails direct contact with the oral mucosa, and many dental treatments produce aerosols [39,66].

Dentists were advised to adapt their infection control routine for these aforementioned reasons. Some important recommendations were [6,7,8]:*Patient screening:*

Information was gathered from the patients about exposure to infected or possibly infected people, travel history, and whether they had some of the most frequent symptoms of COVID-19.


*Evaluation of body temperature of patients:*


Measured on the forehead.


*Hand hygiene:*


Cleaning hands with alcohol-based hand rub or with water and soap. Typically, an alcohol-based hand rub is suggested, but water and soap are preferred if hands are visibly dirty.


*Personal protective equipment:*


Including gown, mask or respirator, goggles or face shields, and gloves.


*Preprocedural mouth rinse:*


Rinses like chlorhexidine or hydrogen peroxide should be used to reduce contaminated aerosols.


*Radiographs:*


If possible, extraoral imaging should be considered to reduce cough or gag reflex.


*Rubber dam:*


Should be used to prevent splatter production.


*Single-use tools:*


Should be used to avoid cross-contamination.


*Reduce aerosol production:*


Manual removal of oral calculus instead of using an ultrasonic scaler, and reducing the usage of high-speed handpieces or three-way syringes.


*Surface disinfection:*


As SARS-CoV-2 is believed to survive up to nine days onto surfaces, surface disinfection is indispensable [7].

The recommended adoptions have not been successfully implemented despite sufficient risk awareness scores. Different countries published various, even contradictory, guidelines [16]. This was challenging for dentists. Dentists should primarily follow the guidelines of their countries’ responsible federal agency. Another reason for low adaption could be the lack of COVID-19 training and information possibilities [20,26,29,34,35,47,52]. More training should be offered, and a broader spread of guidelines should be targeted. Also, exchange opportunities between dentists from the public and private sectors should be organized, as dentists from public sectors were more confident about preventive measures [62]. The majority of dentists wanted to defer treatment of symptomatic or infected people [43,59]; 1–2 out of 10 dentists had treated infected patients so far [23,54]. Dentists also mentioned that they were uncomfortable treating infected people [62], showing that they were aware of the risks. Nonetheless, 78% of dentists thought that treatment should still be offered to COVID-19 positive patients [26]. Transferring infected patients to specialized centers or hospitals could be a solution to this issue [47]. Another idea would be to give dentists better access to recommended personal protective equipment (PPE) and infection control training, so they could also treat infected people.

Even though many dentists were in favor of using N95 masks, the number of dentists who actually used those masks was surprisingly low. Reasons for the low use of N95 masks may be poor availability and the high prices [26]. However, some studies show promising decreases in infection risks compared to ordinary surgical masks [54]. Further research must still be done to answer the question of whether N95 masks are better than regular surgical masks, as there are studies which showed no statistical difference between the two mask types, especially in treatments without aerosol production [67]. It is important too, not to misuse the equipment, as often supply is stated as a problem [67]. Based on the actual study situation, N95 masks can be recommended for aerosol-producing treatments. To improve their usage, national health care authorities could provide these masks to oral health care workers with information on reasonable application.

### 4.1. Strengths of the Study

This research summarizes the findings of previous KAP studies in this field, helping healthcare policymakers to apply this knowledge to disease prevention and control, and also strengthening and improving the health care of dental staff.

### 4.2. Limitation

A key limitation in this study was the lack of a quality assessment tool for the articles included in the systematic review. The majority of papers included in this review were near the bottom of the evidence pyramid, often viewed as having low-quality evidence. Another drawback was the limited number of publications on this subject, particularly from developing and underdeveloped nations, limiting the generalizability of the scoping review results. Another limitation identified in this research was the study’s emphasis on general dentist practices rather than dental specialties such as orthodontics, pediatrics, or geriatrics. Finally, social desirability bias may impair a questionnaire’s validity. Almost all of the research included in this study used questionnaires. Individuals’ tendency to present a favorable picture of themselves on surveys is referred to as a socially desirable response (SDR) [68]. Individuals who score highly on this trait often provide a favorable picture of themselves by refusing to respond honestly, particularly on controversial or sensitive matters [69].

## 5. Conclusions

This review has shown that the dentists’ awareness of COVID-19 symptoms and their overall risk awareness was relatively high; moreover, most dentists in different countries have adopted prescribed preventive measures, but there is still potential to enhance that adoption. It is critical to address the information gap about the complicated nature of COVID-19’s influence on dental practice; there should be more exchange between specialized centers and private practitioners and if possible increased access to PPE.

The current findings may help policymakers understand why it is critical to promote evidence-based medical education for all health care providers, including oral health practitioners. Oral health authorities should prioritize developing thorough recommendations and preventing the spread of misinformation, especially in countries with a high prevalence of infections. More studies should be undertaken to demonstrate whether FFP2 masks are a useful tool for infection protection in dental practice.

## Figures and Tables

**Figure 1 ijerph-19-04971-f001:**
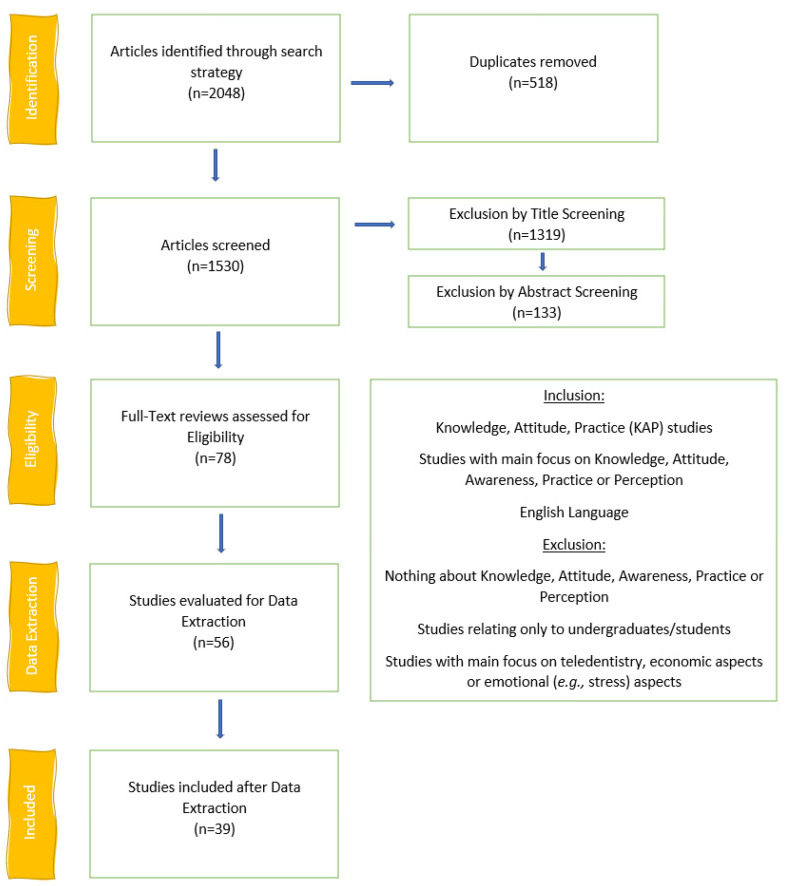
Flow Chart of the search procedure.

**Figure 2 ijerph-19-04971-f002:**
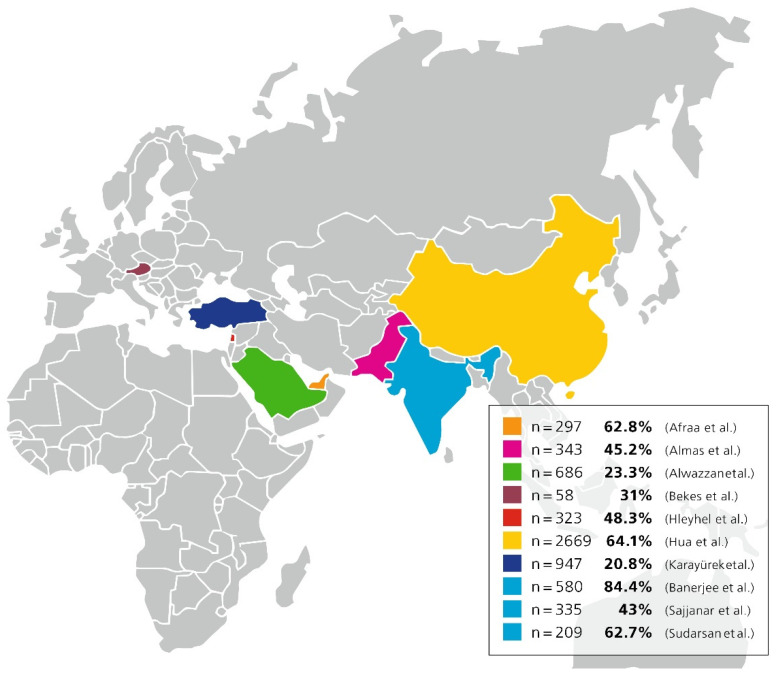
Shows a map of countries with the scores of how many dentists were willing to treat COVID-19 affected subjects in evaluated papers [20,23,29,32,41,47,49,56,57,61].

**Table 1 ijerph-19-04971-t001:** Database search procedure.

Database	Ovid Medline	Scopus	Embase	Medline via PubMED
Search field	Multi-Field-Search	Search	Titel, Abstract, Author, Keywords	Title, Abstract
Search terms	COVID-19, dental, dentistry, dentist	COVID-19, dental, dentistry, dentist	COVID-19, dental, dentistry, dentist	COVID-19, dental, dentistry, dentist
Search constellation	(Dentistry OR dental OR dentist) AND COVID-19	COVID-19 AND dental OR dentistry OR dentist	(Dentist OR dentistry OR dental) AND COVID-19	(Dentist OR dentistry OR dental) AND COVID-19
Results	175	27	670	1176

**Table 2 ijerph-19-04971-t002:** National surveys included sorted by Country, period of the surveys, Relative changes (%) COVID-19 cases and number of dentists participating in the surveys.

		*Our World in Data [19]*		
*Author*	*Country*	*Survey Period*	*Relative Changes* (%)	*Participants* (*n*)
Bekes et al. [20]	Austria	June 2020	−12	58
Gomes et al. [21]	Brazil	May 2020	+44	1105
Rossato et al. [22]	Brazil	June–July 2020	+174	1357
Hua et al. [23]	China	April 2020	8	2669
Önöral et al. [24]	Cyprus	April–May 2020	−33	228
Widyarman et al. [25]	Indonesia	June 2020	+2	632
Alduwayhi et al. [26]	India	July–August 2020	+205	459
Arif et al. [27]	India	April 2020	+860	302
Srivastava et al. [28]	India	April 2020	+1243	311
Banerjee et al. [29]	India	May 2020	+2028	493
Izna et al. [30]	India	July 2020	+79	124
Kanaparthi et al. [31]	India	July 2020	+79	385
Sajjanar et al. [32]	India	July 2020	+79	355
Ravi et al. [33]	India	May 2020	+2028	825
De Stefani et al. [34]	Italy	April 2020	+21	1500
Cagetti et al. [35]	Italy	April 2020	+21	3599
Gambarini et al. [36]	Italy	April 2020	+21	700
Putrino et al. [37]	Italy	February–March 2020	+91	535
Sinjari et al. [38]	Italy	March–May 2020	+53	1185
Khader et al. [39]	Jordan	May–June 2020	+15	368
Nasser et al. [40]	Lebanon	April–May 2020	--	358
Hleyhel et al. [41]	Lebanon	May–August, 2020	--	323
Santana et al. [42]	Mexico	June 2020	+228	336
Gómez-Clavel [43]	Mexico	May–July 2020	+43	703
Pandey et al. [44]	Nepal	June–August 2020	+584	384
Imran et al. [45]	Pakistan	July 2020	−9	822
Chaudhary et al. [46]	Pakistan	March–June 2020	+16	583
Almas et al. [47]	Pakistan	April–May 2020	9	343
Khan et al. [48]	Pakistan	May-June 2020	+15	306
Alwazzan et al. [49]	Saudi Arabia	July 2020	+35	686
Srivastava et al. [50]	Saudi Arabia	March 2020	+96	318
Mustafa et al. [51]	Saudi Arabia	March 2020	+96	269
Al-Khalifa et al. [52]	Saudi Arabia	May 2020	+14	287
Martínez-Beneyto et al. [53]	Spain	April 2020	−49	4298
Wolf et al. [54]	Switzerland	July–August 2020	+195	1324
Tokuç et Coskunses [55]	Turkey	March–April 2020	+13	590
Karayürek et al. [56]	Turkey	July–August 2020	+110	947
Afraa et al. [57]	UAE	June 2020	−32	297
** *Total* **				**30,364**

**Table 3 ijerph-19-04971-t003:** Awareness percentage of symptoms in the included studies.

*Author*	*Country*	*Awareness of Symptoms* (%)
Bekes et al. [20]	Austria	84.5
Alduwayhi et al. [26]	India	99.0
Arif et al. [27]	India	98.7
Srivastava et al. [28]	India	88.0
De Stefani et al. [34]	Italy	93.7
Cagetti et al. [35]	Italy	93.4
Khader et al. [39]	Jordan	91.8
Nasser et al. [40]	Lebanon	93.9
Santana et al. [42]	Mexico	99.4
Alwazzan et al. [49]	Saudi Arabia	96.9
Srivastava et al. [50]	Saudi Arabia	97.5
Tokuç et Coskunses [55]	Turkey	96.3
* **Mean (±Standard deviation)** *		**94.4 ± 4.8**

**Table 4 ijerph-19-04971-t004:** Shows the risk awareness of dentists.

Author	Risk Awareness
Ahmed et al. [59]	86% expressed fear after learning about COVID-19-related fatalities. A significant proportion of dentists (66%) expressed a desire to stop their clinics until the number of COVID-19 cases began to decline.
Alduwayhi et al. [26]	80.7% of dentists feel coronavirus is fatal, and 99% believe they are at higher risk.
Alwazzan et al. [49]	The virus was assessed as dangerous by 90.2% of dentists, and 96.1% believe it is a significant public health concern. More experienced and male dentists had greater levels of awareness.
Banerjee et al. [29]	Among respondent dentists 87% fall into the very high-risk group, 58 (10%) indicated that they belong to the high-risk category, and 17 (3%) stated that they belonged into the low-risk category.
Bekes et al. [20]	Over two-thirds (67.2%) rated the infection as fairly dangerous in general, while one-half (55.2%) assessed the chance of catching the infection as moderately dangerous.
Chaudhary et al. [46]	The odds of having awareness about the risk of exposure to COVID-19 and fear of getting infected were greater in clinical than non-clinical OHCW (OR: 52.6; OR: 15.9, respectively).
Imran et al. [45]	In general, most of the participants were aware of the COVID-19 pandemic and concerned about the risk to their families.
De Stefani et al. [34]	Becoming personally infected is moderately dangerous.
Gambarini et al. [60]	While 54% of dentists thought there was a higher risk for the patient and 43% thought the risk higher for dental professionals, only 3% evaluated the risk as equal.
Gambarini et al. [36]	Overall, 70% of dentists believed that dental settings are more dangerous for the spread of COVID-19 than other social behaviors. The majority of respondents believed that dentists faced the greatest risks, while only 5% believed patients faced the greatest risks; a significant number of respondents believed risks were evenly distributed between patients and dentists.
Gomes et al. [21]	A total of 1011 (91.5%) respondents identified high risk of COVID-19 infection to dentists.
Gomez-Clavel et al. [43]	Risk awareness score was 97.2%
Hleyhel et al. [41]	While 18.6% of the respondents reported perceiving COVID-19 as very dangerous, 46.4% and 35% reported perceiving it as moderately dangerous and not dangerous.
Hua et al. [23]	COVID-19 risk to patients and HCWs (84.1%)
Imran et al. [45]	Most dental practitioners (71%) agreed that they are at greatest risk of coronavirus infection due to their close contact with infected patients and aerosol inhalation.
Izna et al. [30]	Numerous dentists (80.6%) were aware of having the greatest risk of catching COVID-19 infections of any profession.
Kamate et al. [58]	COVID-19 was thought to be deadly by 95.9% of dentists, and 99.8% stated that using face masks may prevent transmission.
Kanaparthi et al. [31]	Overall, 93.5% were aware that dentists were at risk.
Karayürek et al. [56]	When asked whether dental procedures may increase COVID-19 infections, 40.9% answered yes. Older dentists thought dental procedures could not raise COVID-19 infection risk. Although specialized dentists performed fewer dental examinations than general dentists during the pandemic, they believed dental treatments increased COVID-19 infection risk.
Mustafa et al. [51]	Participants in the age groups “over 60”, “50–59”, and “20–29” were more likely to perceive it as a very dangerous disease (80.0%, 80.4%, and 65.8% of participants, respectively) compared to the 30–39 and 40–49 age groups (45.2% and 48.8% of participants, respectively)

**Table 5 ijerph-19-04971-t005:** Preventive measures scores.

*Author*	*Country*	*Participants* (*n*)	*Preventive Measures* (%)
Afraa et al. [57]	UAE	297	95.7
Almas et al. [47]	Pakistan	343	54.8
Alwazzan et al. [49]	Saudi Arabia	686	83.1
Hleyhel et al. [41]	Lebanon	323	71.2
Hua et al. [23]	China	2669	76.1
Kamate et al. [58]	Asia	260	98.5
Kamate et al. [58]	Americas (North and South)	212	98.6
Kamate et al. [58]	Europe	139	99.3
Kamate et al. [58]	Africa	193	99.5
Kamate et al. [58]	Other continents (Australia and Antarctica)	46	97.9
Nasser et al. [40]	Lebanon	358	96.4
Putrino et al. [37]	Italy	535	69
Rossato et al. [22]	Brazil	1178	98
Widyarman et al. [25]	Indonesia	632	96
* **Total** *		**7871**	
* **Mean (±Standard deviation)** *			**88.2 ± 14.6**

**Table 6 ijerph-19-04971-t006:** Recommendation and use of N95 masks expressed as percentage of dentists.

			N95	
Author	Countries	Participants (*n*)	Recommendation (%)	Use (%)
Ahmed et al. [59]	Australia, Bahrain, Bulgaria, Canada, China, Egypt, Finland, France, Germany, Hungary, India, Ireland, Israel, Italy, Kuwait, Malaysia, Mexico, New Zealand, Pakistan, Poland, Republic of Congo, Romania, Saudi Arabia, South Africa, Switzerland, Turkey, United Arab Emirates, and the United States of America	650	84	N/A
Alduwayhi et al. [26]	India	459	79	10
Al-Khalifa et al. [52]	Saudia Arabia	287	72	N/A
Bakaeen et al. [62]	Eastern Mediterranean	347	63.2	34.4
Bakaeen et al. [62]	Europe	449	41.6	8.3
Bakaeen et al. [62]	North America	234	67.8	16.4
Bakaeen et al. [62]	Western Pacific	158	40.5	25.3
Bekes et al. [20]	Austria	58	N/A	89.7
Kanaparthi et al. [31]	India	358	94.8	N/A
Karayürek et al. [56]	Turkey	947	N/A	47.6
Khan et al. [48]	Pakistan	306	89.5	N/A
Martinez-Beneyto et al. [53]	Spain	6470	25.3	N/A
Nasser et al. [40]	Lebanon	358	80	N/A
Pandey et al. [44]	Nepal	384	81.7	N/A
Ravi et al. [33]	India	808	95	N/A
Casillas Santana et al. [42]	Mexico	339	N/A	70.2
** *Total* **		**12,612**		
** *Mean (±Standard deviation)* **			**70.3 ± 22.1**	**37.7 ± 27.6**

## Data Availability

No new data were created or analyzed in this study. Data sharing is not applicable to this article.

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
