# Peer review of "How the COVID-19 Pandemic Affects Risk Awareness in Dentists: A Scoping Review"

_ijerph, 2022, doi:10.3390/ijerph19094971_

Round 1

Reviewer 1 Report

Manuscript entitled “How the COVID-19 pandemic affects risk awareness and risk perceptions in dentists: A Scoping Review” is review paper that analyzes 39 scientific papers showing the results of research of the dentists' risk awareness, awareness of the COVID-19 symptoms, preventative measures, and effective methods of preventing and managing the COVID-19 infection, as stated.

 The title maybe could be improved because terms risk awareness and risk perception are quite close and could be difficult for reader to distinguish what is meant by them.

Second sentence in Material and methods section is strange, should be rewritten (lines 92-95).

I have notice terms preventative measures and preventive measures, both in different places through text, would be better to chose only one term for consistency.

PPE abbreviation is not explained and was used more than once.

The study itself is good and manuscript is written sound, with all limitations and strength clearly indicated.

Reference list has certain shortcomings (references 5, 19, 66,....) so it needs detailed review and correction of errors.

Generally, the manuscript is acceptable for publication in this journal after minor corrections.

Reviewer 2 Report

This manuscript presents the result of documentary reviews on the dentists' risk awareness, awareness of the COVID-19 symptoms, preventative measures, and effective methods of preventing and managing the COVID-19 infection. Studies related to the dentists' risk awareness and relevant issues were analyzed. Those studies were conducted in several countries such as Saudi Arabia, India, Turkey, Lebanon, etc. The idea of this this study is interesting, but I disagree with using documentary analysis method for this research. The studies employed for data analysis in this study were conducted in several countries where experienced a different degree of COVID-19 outbreak. Therefore, it  is meaningless to compare risk perception of dentists from different countries without considering actual COVID-19 outbreak in each country. Based on my opinion, this paper still contains several weaknesses, and it must be substantially improved for the publication in an academic paper. I have following comments;

1.the contribution of this study is not clear. Since each country has faced a different situation of COVID-19 outbreak, the dentists' risk awareness and the degree of adoption of preventative measures must be absolutely different. The result is not surprising for me. 

2. The authors borrowed several studies conducted in many countries. The details of those studies were not presented. Those details are such as analysis method, survey method, scale of survey, COVID-19 outbreak situation etc. The results of those studies such as risk awareness can be comparable, if those studies used the same methods. Therefore, the result is not convincing for me. For instance, page 3 line 36, "About three-quarters of dentists ranked the virus as moderately dangerous or dangerous [18, 20, 21, 31, 32].  Did those studies measure risk perception in the same ways. 

3.This study did not clearly present how risk awareness, awareness of symptoms, preventive measures were defined.

4. It would be more appropriate to explain details of preventive measures for the dentists.

5. Regarding treatment of COVID19 affected subjects, I am curious authors explored actual practices or their willingness?

6. For discussion part, the result should be discussed based on COVID-19 spreading situation in each country , and it would be more appropriate to present practical contributions or theoretical contributions of this study. 

Reviewer 3 Report

Figure 1. "Data Extraction..." Please include a list with the variables of extraction. For example:   3 studies were extracted due to no relationship with the clinical history. 5 studies were extracted due to...... Table 3. Were only these authors included preventive measures? Or why were only these authors described in this table?   Figure 2. This is not a world map, only Africa, Asia, and Europe were described. Would it be possible to describe American Continent? Otherwise, I suggest to change the title.   Table 4. "Countries..." The order of countries was established according to the author or percentages of use, if not, would it be possible to use alphabetical order? Material and Methods Would it be possible to describe the date when the data were obtained? For example: The manuscripts were selected from 1st March 2021 to 1st March 2022. Lines 222 to 241. Please explain how these results were obtained. Please explain the percentage of manuscripts in which these variables were reviewed.   Line 243. "As SARS-CoV-2...", please cite these lines.  Line 246. "Different countries..."  Why were these events contradictory? Please provide a reference or explain the most common reported events.

Round 2

Reviewer 2 Report

 The revised version was partially improved. Some important comments were not yet addressed properly.

  1. Could you please clearly indicate the contribution of this research? Why is it significant to know the dentists' perceived risk related to COVID-19 outbreak in several countries and their adoption of preventative measures. Why is it important to compare risk perception among dentists from several countries which have experienced a different degree of COVID-19 situation? Do you want to raise their awareness or do you want to enhance their preventive measures?. Authors just indicated that preventive measures taken by dentists in those countries can be learnt from this study. If this is only a contribution, this study can only explore effective preventive measures taken by the dentists. It is no need to know and compare risk awareness and a degree of adoption of preventative measures among dentists in several countries.
  2. For instance, if the result reveals that a great proportion of dentists in a country has a low risk perception and less adopts preventive measures. Whereas, that country has faced a serious COVID-19 situation (high infection rate and fatality rate).Therefore, it could be recommended that effective communication among the dentists must be established in that country. Therefore, I think that the authors should discuss the results by trying to explain reasons behind the results, and propose contributions. In this sense, COVID-19 outbreak situation (such as infection rate and fatality rate) in those studied countries should be used to discuss the results. I think authors still need to improve the discussion part.
  3. In another case, a number/proportion of dentists who adopted preventative measures were different in each country. Some countries such as Pakistan and Italy had a lower number of dentists adopting preventative measures than other countries.  Is that because of the low severity of COVID-19 situation experienced by those countries or the insufficient knowledge of the dentists. or other reasons? A low level of adoption of preventative measures by the dentists can be problematic, if COVID-19 spreading situation is serious. Authors can propose a recommendation to enhance a level of adoption of the measures. This is why I recommended authors to explore and indicate COVID situation in those studied countries as well.
  4. A degree of individual risk awareness and risk perception can be determined by some significant factors such as individuals' perceived exposure, perceived severity of consequences, and perceived probability of a negative event's occurrence. If this is well understood, explanations on the difference in risk awareness and adoption of COVID-19 prevention measures among dentists in each country can be explained.
  5. Regarding the discussion, authors indicated that 

    "The aim of this paper was to report a holistic look at how dentists generally assessed the risk of the pandemic in different parts of the world." 

     Of course, it can be done, if it is published in a general report. If it would be published in an academic paper, scientific explanations and contribution must be presented. This means all results must be scientifically discussed, and concrete contributions must be presented.
